# Experiments and 3D Molecular Model Construction of Lignite under Different Modification Treatment

**Jihui Liu [1], Shuang Qiu [1], Zhijun He [1,\*] and Yaowei Yu [2,\*]**

[1] School of Materials and Metallurgy, University of Science and Technology Liaoning, Anshan 114051, China; gtyj66@126.com (J.L.); daiwha_baosteel@163.com (S.Q.)
[2] State Key Laboratory of Advanced Special Steel, Shanghai Key Laboratory of Advanced Ferrometallurgy, School of Materials Science and Engineering, Shanghai University, Shanghai 201900, China
\* Correspondence: hezhijun@ustl.edu.cn (Z.H.); Yaowei.yu@Hotmail.com (Y.Y.)

**Abstract:** In this paper, Huolinhe lignite was selected as the lignite experimental sample, using microwave modification and ultrasonic modification separately as improvement methods. The three-dimensional molecular models of HLH before and after modification were established base on the parameters obtained by $^{13}$C NMR, X-ray photoelectron spectroscopy (XPS), Raman spectroscopy (Raman), and Fourier transform infrared (FTIR). After the microwave treatment, the methylene carbon in the HLH coal sample structure mostly exists in the form of long straight chains, and after microwave and ultrasonic treatment, the -OH content of oxygen atoms in the coal sample increases, and form the CO- and the COO-. The proportion is decreasing. The models were adjusted and tested by the covalent bond concentration method and carbon chemical shift spectra calculation using Chemdraw software. A new method is proposed to study the structure and physicochemical properties of lignite modification from the molecular point of view through this study.

**Keywords:** Lignite; microwave and ultrasound modification; structural characterization; 3D molecular model; structural simulation

---

## 1. Introduction

In recent years, the shortage of high-rank coal resources has gradually become a prominent problem in industrial development [1,2]. Lignite is widely used in energy fields, such as pyrolysis, combustion, gasification and liquefaction [3,4]. In order to make more effective use of lignite resources, many scholars carried out much research on modification treatment processes for lignite characteristics. Arash Tahmasebi [5] discovered that the content of some functional groups in pulverized coal particles decreased significantly after microwave irradiation, but the content of aromatic carbon and aromatic ring in lignite was not affected by microwave pyrolysis. Sun Qiang [6] selected coal samples were treated with water and heat treatment and found that the rate of re-absorption decreased with the increase of temperature, and the lignite quality could improve most in high temperature and low humidity. Ge Lichao [7] found the rank of lignite increased after microwave modification and the combustion reaction process moved to high temperature zone by Thermogravimetry (TG) analysis.

The existing research focuses more on the optimization of modification processes and proposes new modification processes. The mechanism of these processes were difficult to study by experimental methods due to innumerable coupling reaction pathways during the utilization of lignite [8–10]. Therefore, Huolinhe lignite (HLH) was selected as the experimental sample, using microwave modification (MM) and ultrasonic modification (UM) as improvement methods separately. The two-dimensional molecular models of HLH before and after modification were established based on the parameters obtained by a series of detection methods, and three-dimensional model is constructed

based on molecular mechanics and molecular dynamics. A new method is proposed to study the structure and physicochemical properties of lignite modification from the molecular point of view through this study.

## 2. Experiment

### 2.1. Modification of Lignite

HLH was low degree of coalification was selected as the experimental sample. HLH sample was ground to 109–180 μm and dried in a vacuum drying chamber at 40 °C for 24 h. Take HLH sample in a crucible, added 100 mL distilled water and blended fully. The crucible contained HLH sample was placed in the ultrasonic oscillator to water bath oscillation for 60 s. The crucible was placed in the drying oven for drying treatment with 85 °C for 4 h. The crucible containing the HLH sample was placed in the a quartz reaction tube of the microwave reactor, the modification parameters set as 200 W and 60 s, and the microwave activated. Industrial analysis and elemental analysis of lignite samples before and after treatment were carried out and the results are shown in Table 1.

**Table 1.** Proximate and ultimate analyses of HLH lignite.

| Sample (wt. %), ad | Proximate Analysis | | | | Ultimate Analysis | | | | |
|---|---|---|---|---|---|---|---|---|---|
| | $M_{ad}$ | $A_{ad}$ | $FC_{ad}$ | $V_{daf}$ | C | H | O | N | S |
| HLH | 16.32 | 21.97 | 24.20 | 37.51 | 78.38 | 6.24 | 13.35 | 1.51 | 0.52 |
| MM | 15.22 | 22.32 | 26.03 | 36.43 | 80.67 | 5.81 | 11.29 | 1.66 | 0.57 |
| UM | 15.46 | 22.64 | 25.06 | 36.84 | 81.77 | 5.47 | 10.31 | 1.85 | 0.60 |

Note: ad: air-dry basis; daf: dry-and-ash-free basis. M: moisture; A: ash; FC is fixed carbon; V: volatile matter content.

### 2.2. FTIR and Structural Parameters Analysis

#### 2.2.1. FTIR Results Analysis

Infrared spectroscopy is closely related to the chemical structure of the substance. It can be confirmed the aromatic structure, oxygen-containing structure and fat structure of coal by FTIR detection [11]. Infrared spectra of all samples are shown in Figure 1, with curves smoothed and baselines corrected.

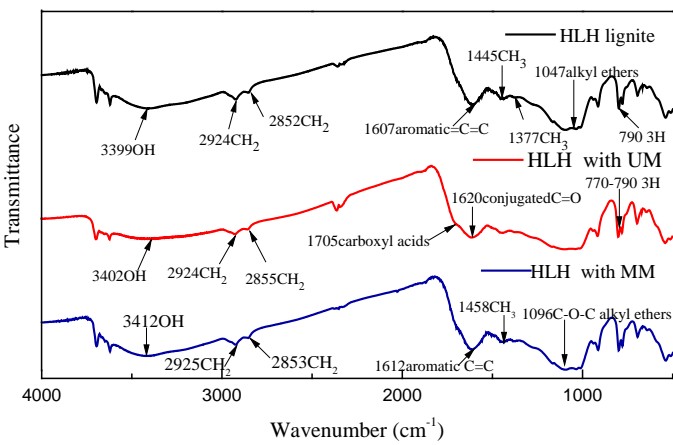

**Figure 1.** FTIR spectra of lignite before and after modification.

It can be observed that selected coal samples contain similar functional groups, hence the absorption of sample to infrared spectrum occurs at same wavenumber positions [11]. These obtained spectra are comprehensive curves of many independent peaks, which have to be deconvoluted to achieve. All spectra were divided into 4 regions, namely 700–900 cm$^{-1}$ (aromatic structures), 1000–1800 cm$^{-1}$

(oxygen-containing structures), 2800–3000 cm$^{-1}$ (aliphatic structures) and 3000–3600 cm$^{-1}$ (hydroxyl structure). Consequently, the area values of specific peak can be derived. The corresponding relationships between peak position and functional group are shown in Tables 2–5 [12,13].

**Table 2.** Aromatic structure of HLH before and after modification.

| Assignment | Relative Area | | |
|---|---|---|---|
| | HLH | MM | UM |
| 4H | 11.051 | 13.573 | 12.106 |
| 3H | 86.498 | 86.427 | 87.894 |
| 2H | 2.459 | — | — |

**Table 3.** Oxygen-containing functional group of HLH before and after modification.

| Assignment | Relative Area | | |
|---|---|---|---|
| | HLH | MM | UM |
| Alkyl ethers | 16.406 | 14.231 | 13.321 |
| C-O phenols, ethers | 48.679 | 40.349 | 51.372 |
| C-O in aryl ethers | 9.023 | 11.452 | 8.213 |
| Symmertric CH$_3$-Ar, R | 1.612 | 1.823 | — |
| Asymmertric CH$_3$-, CH$_2$- | 2.462 | 7.435 | 9.468 |
| Aromatic C=C | 10.954 | 9.097 | 6.039 |
| Conjugated C=O | 9.633 | 10.979 | 10.534 |
| Carboxyl acids | 1.231 | 4.634 | 1.053 |

**Table 4.** Fatty structure of HLH before and after modification.

| Assignment | Relative Area | | |
|---|---|---|---|
| | HLH | MM | UM |
| Sym. R$_2$CH$_2$ | 13.118 | 20.622 | 17.705 |
| R$_3$CH | 47.68 | 28.051 | 36.046 |
| Asym. R$_2$CH$_2$ | 24.814 | 30.553 | 27.132 |
| Asym. RCH$_3$ | 14.388 | 20.774 | 19.118 |

**Table 5.** Hydroxyl structure of HLH before and after modification.

| Assignment | Relative Area | | |
|---|---|---|---|
| | HLH | MM | UM |
| OH-N | 4.419 | 3.118 | 8.304 |
| Ring hydroxyl | 45.653 | 41.322 | 31.922 |
| Phenol OH | 35.271 | 41.202 | 42.944 |
| OH-$\pi$ | 14.662 | 14.358 | 16.829 |

It can be found that there are 3 substitution modes of hydrogen atoms on benzene ring in HLH structure. The proportion of triple substituted aromatics (3H) is the largest among all the samples. The proportion of triple substituted aromatics of HLH raw coal is 86.498%, the MM sample is 86.427% in, the UM sample is 87.893%. The tetrasubstituted hydrocarbons (2H) of HLH lignite has also changed significantly. Tetrasubstituted hydrocarbons (2H) are not found in the infrared spectra of HLH lignite after both modification. In the process of modification, the substitution reaction of aromatic hydrocarbons may cause structural changes, which is due to the instability of other atoms and aliphatic side chains in benzene rings.

It is also found that the form and proportion of oxygen elements in HLH sample changed a lot after modification, however the carbonyl (C=O), alkyl ether (C-O-C), and phenol hydroxyl (-OH) groups are still the main existing forms of oxygen-containing functional group of HLH.

### 2.2.2. FTIR Structural Parameters Analysis

FTIR structural parameters could be obtained by the area of peak with peak fitting [14,15].
(1) Ratio of hydrogen to carbon H/C.

$$\frac{H}{C} = \frac{H_{ad}}{\frac{C_{ad}}{12}} \tag{1}$$

(2) The aromatic carbon ratio $f_{ar-F}$: On the premise of ignoring carbonyl carbon, assuming that coal only contains aromatic carbon and aliphatic carbon, the formula for calculating aromatic carbon rate is as follows:

$$\frac{H_{al}}{H} = \frac{A(3000-2800)\text{cm}^{-1}}{A(3000-2800)\text{cm}^{-1} + A(900-700)\text{cm}^{-1}} \tag{2}$$

$$f_{ar-F} = 1 - \frac{C_{al}}{C} = 1 - \frac{\frac{H_{al}}{H} \times \frac{H}{C}}{\frac{H_{al}}{C_{al}}} \tag{3}$$

where aromatic hydrogen ratio $H_{ar}/H$: $C_{al}/C$ is the ratio of aliphatic carbons to the total number of carbons, H/C represents the ratio of hydrogen to carbon atoms, $H_{al}/C_{al}$ is 1.8 for all coal samples, and represents the atomic ratio between hydrogen and carbon in aliphatic groups.
(3) Fat carbon ratio $f_{al-F}$:

$$f_{al-F} = 100 - f_{ar-F} \tag{4}$$

(4) Lipid chain length and branching degree of coal $I_1$: According to the ratio between $CH_2$ and $CH_3$, namely the area ratio of $A(CH_2)/A(CH_3)$, the aliphatic group length and the branched chain degree were calculated to determine the aliphatic structural parameters. The intensity ratio of $CH_2/CH_3$ was determined by Equation (5):

$$I_1 = \frac{CH_2}{CH_3} = \frac{A_{2852\text{cm}^{-1}} + A_{2924\text{cm}^{-1}}}{A_{2957\text{cm}^{-1}}} \tag{5}$$

(5) Alkane branching degree:

$$\delta_F > \frac{R_3CH}{A_{(3000-2800\text{cm}^{-1})}} \tag{6}$$

Compared with the alkane branching degree of the 3 coal samples from Table 6, the $\delta_F$ of HLH without modification is 47.68% which indicate there were much tertiary carbon and quaternary carbon in HLH raw coal. There were many branching structures in HLH raw coal. After microwave modification, the main structure of HLH is methylene carbon with long straight chain.

**Table 6.** FTIR structural parameters of HLH before and after modification.

| Parameter | HLH | MM | UM |
|---|---|---|---|
| H/C | 0.97 | 0.89 | 0.83 |
| $H_{al}/H$ | 0.56 | 0.44 | 0.45 |
| $f_{ar-F}$ | 61.38 | 78.49 | 79.84 |
| $f_{al-F}$ | 38.62 | 21.52 | 16.16 |
| $I_1$ | 2.64 | 2.85 | 2.35 |
| $\delta_F$ | 47.68% | 25.05% | 36.05% |

### 2.3. $^{13}C$ NMR Results Analysis

Figure 2 shows $^{13}C$ NMR spectra of HLH lignite. It is including 2 main peaks. The lipid-carbon peak area with chemical shift of $0–90 \times 10^{-6}$ and the aromatic-carbon peak area with chemical shift of $90–165 \times 10^{-6}$. The sample also contains a small amount of carbonyl carbon, with a chemical shift of $165–220 \times 10^{-6}$ in the peak area [16,17]. The carbon spectra obtained before and after modification were fitted by peak-splitting method, and 9 carbon skeleton structural parameters were obtained. The results are shown in Table 7.

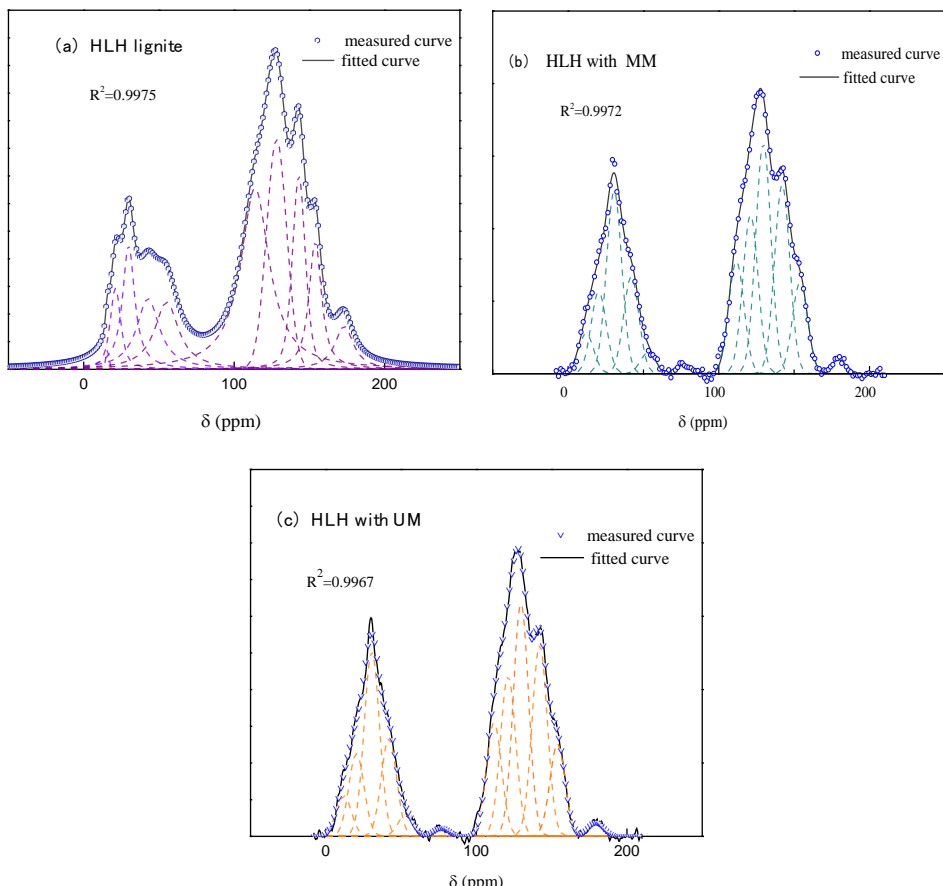

**Figure 2.** Peak fit of $^{13}C$ NMR of HLH before and after modification.

**Table 7.** Structure attribution and relative content of chemical shifts in $^{13}C$ NMR spectra of HLH before and after modification.

| Chemical Shift | Structural Fragments | Symbols | Carbon Distribution | Sample | | |
|---|---|---|---|---|---|---|
| | | | | HLH | MM | UM |
| 14–16 | Aliphatic CH$_3$ | $f_{al}^M$ | | | | |
| 16–22 | Aromatic CH$_3$ | $f_{al}^A$ | Aliphatic carbon | 30.48 | 32.246 | 32.21 |
| 22–50 | Methylene | $f_{al}^H$ | | | | |
| 50–90 | Oxy- aliphatic carbon | $f_{al}^O$ | | | | |
| 100–129 | Aromatic protonated | $f_a^H$ | | | | |
| 129–137 | Aromatic bridgehead | $f_a^B$ | Aromatic carbon | 64.983 | 63.719 | 64.703 |
| 137–148 | Aromatic branched | $f_a^S$ | | | | |
| 148–165 | Oxy- aromatic carbon | $f_a^O$ | | | | |
| >165 | Carbonyl carbon | $f_a^C$ | Carbonyl carbon | 4.319 | 3.135 | 3.087 |

$X_{BP} = fa^B / (fa^H + fa^O + fa^S + fa^B)$, the ratio of aromatic bridge carbon to peripheral carbon of HLH before and after modification is an important parameter to construct macromolecular structure model of lignite, which represent condensation degree of polycyclic aromatic hydrocarbons as well as reflecting the size of aromatic cluster. According the parameters shown above, the $X_{BP}$ of 3 samples could be calculated, and the value of HLH is 0.26, 0.29 for sample with microwave modification, and 0.28 for sample with ultrasonic modification.

### 2.4. XPS Results Analysis

#### 2.4.1. Oxygen Element Analysis

XPS is often used to characterize the existence of oxygen, nitrogen and other heteroatoms in coal, which provides an important basis for the construction of macromolecular structure model of HLH [18]. XPS tests of HLH raw coal, microwave modified HLH and ultrasonic modified HLH were carried out, and the XPS maps of 3 samples were processed by peak fitting. The peak-splitting diagram is shown in Figure 3, and the form and content of nitrogen elements are shown in Table 8.

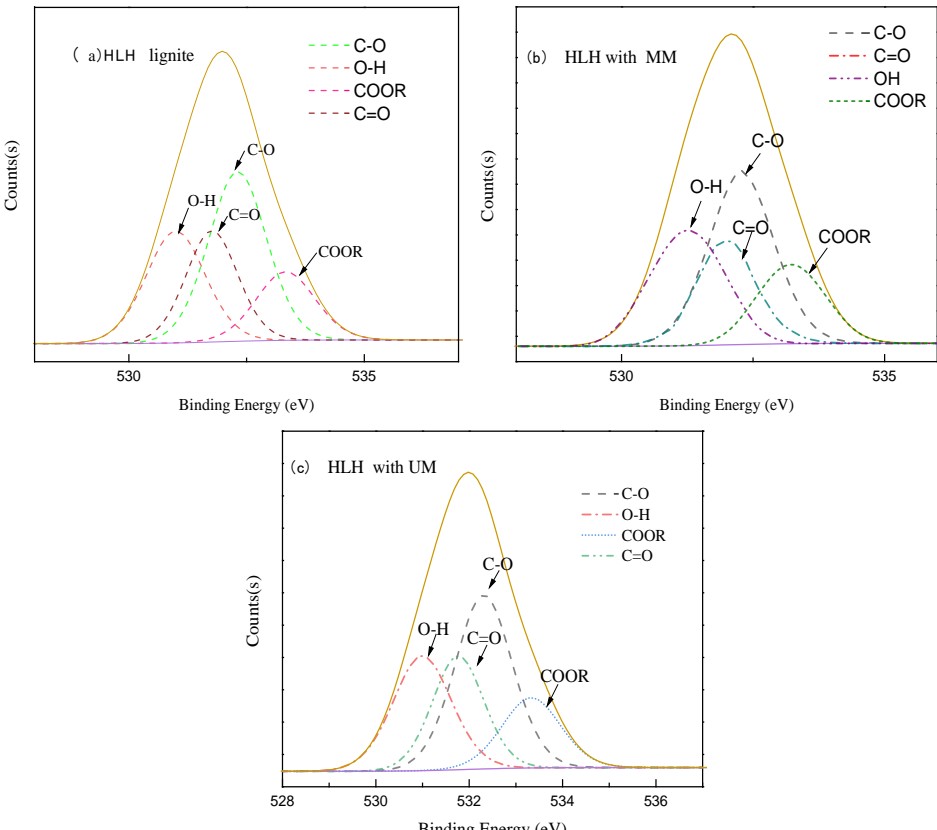

**Figure 3.** XPS peak fitting of oxygen atoms of HLH before and after modification.

The main forms of oxygen in HLH coal samples are C=O, C-O, -OH and COO-, and it can be found in Table 8 that oxygen exists in most of the four forms of C-O in structure. The content of C-O in HLH samples after microwave modification decreases 2%, but the content of COO-structure form decreases from 14.37% to 35.25% of raw coal. 12.37%. The reason for this change may be that microwave treatment destroys the oxygen structure of HLH raw coal. The content of oxygen-OH and C=O in HLH coal samples treated by ultrasonic wave also increased obviously, while the content of CO-form in opposite direction was still decreasing. Therefore, the microwave and ultrasonic treatment of HLH has different effects on the existing forms of oxygen elements. The stability of COO-and C-O structure forms is relatively poor and easy to be destroyed in the process of modification. On the

contrary, the stability of C=O structure makes it not easy to be destroyed in the process of ultrasonic and microwave treatment, which increases the proportion of C=O in the structure.

**Table 8.** XPS detection and analysis of oxygen composition forms of HLH before and after modification.

| Elemental Peak | Functionality | Binding Energy (eV) | Molar Content (%) |
|---|---|---|---|
| HLH | C-O | 532.87 | 37.25 |
| | C=O | 531.27 | 24.71 |
| | -OH | 529.89 | 23.67 |
| | COO- | 533.28 | 14.37 |
| MM | C-O | 532.08 | 35.25 |
| | C=O | 531.87 | 26.47 |
| | -OH | 530.08 | 25.91 |
| | COO- | 533.28 | 12.37 |
| UM | C-O | 532.08 | 36.56 |
| | C=O | 531.87 | 26.21 |
| | -OH | 530.08 | 25.35 |
| | COO- | 533.28 | 11.88 |

### 2.4.2. Nitrogen Element Analysis

Nitrogen in coal mainly comes from coal-forming plants, and most of it exists in the form of organic matter, which mainly includes pyridine nitrogen (N-6), pyrrole nitrogen (N-5), nitrogen oxides (N-X) and proton nitrogen (N-Q) [19,20]. In order to characterize the forms of nitrogen elements in HLH before and after modification, the XPS spectra of HLH lignite were fitted by peak-splitting method. The peak-splitting diagram is shown in Figure 4, and the forms and contents of nitrogen elements are shown in Table 9.

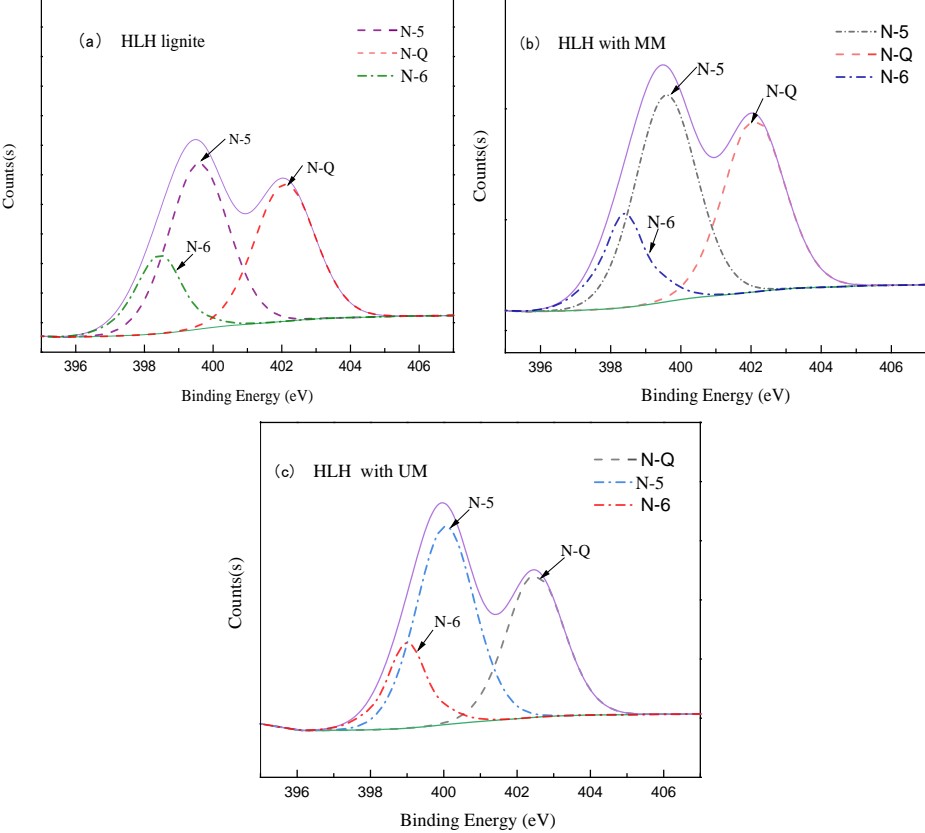

**Figure 4.** XPS peak fitting of nitrogen atoms of HLH before and after modification.

**Table 9.** XPS detection and analysis of nitrogen composition forms of HLH before and after modification.

| Elemental Peak | Functionaliy | Binding Energy (eV) | Molar Content (%) |
|---|---|---|---|
| HLH | N-Q | 402.21 | 32.78 |
| | N-5 | 396.75 | 42.81 |
| | N-6 | 399.81 | 24.41 |
| MM | N-Q | 402.21 | 31.28 |
| | N-5 | 396.75 | 43.02 |
| | N-6 | 399.81 | 25.70 |
| UM | N-Q | 402.21 | 30.78 |
| | N-5 | 396.75 | 43.31 |
| | N-6 | 399.81 | 25.91 |

By comparing the XPS data of N element in HLH before and after modification, it is obvious that the percentages of N-5, N-6 and N-Q in HLH have changed significantly. The percentage of N-Q in pulverized coal decreased to 31.28% and 30.78% respectively after modification, and the corresponding proportion of N-5 and N-6 increased to varying degrees. The total amount of N-5 and N-6 of modified HLH is nearly 70%. It can be seen that microwave and ultrasonic modification methods mainly play a role in N-Q, while N-5 and N-6 form of nitrogen bond are relatively stable, and the above modification methods have little effect on it. Therefore, in order to make the model representative, N-5 and N-6 nitrogen bonds are often used in the construction of macromolecule HLH lignite structure model.

*2.5. Raman Results Analysis*

There are two relatively broad D and G peaks in the Raman spectra of HLH, and two vibration peaks have abundant information, the G peak in lignite does not really represent its crystal structure. It mainly reflects the strength of the stretching vibration bond of aromatic rings [21,22]. Raman spectra of coal samples are exhibited in Figure 5.

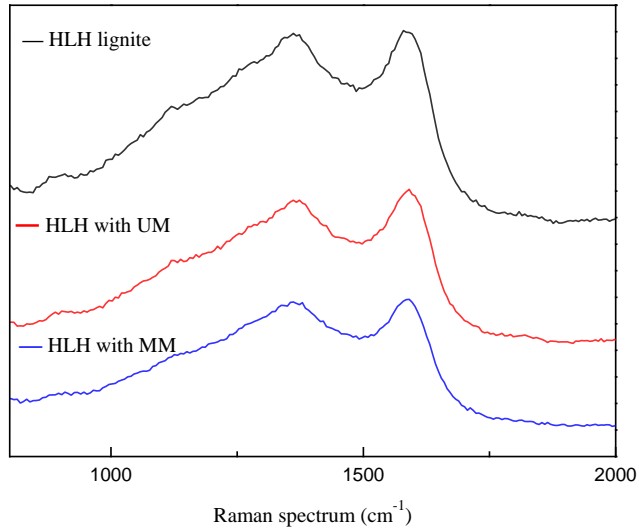

**Figure 5.** Raman spectra of HLH coal samples before and after modification.

There are many overlapping peaks between D and G peaks in Raman spectra of coal. In order to obtain more accurate functional group information of HLH lignite before and after modification, Position and area values of D and G peaks of coal samples obtained by deconvolution process of Raman spectra shown in Figure 6. Raman fitting parameters of HLH coal samples before and after modification are shown in Table 10.

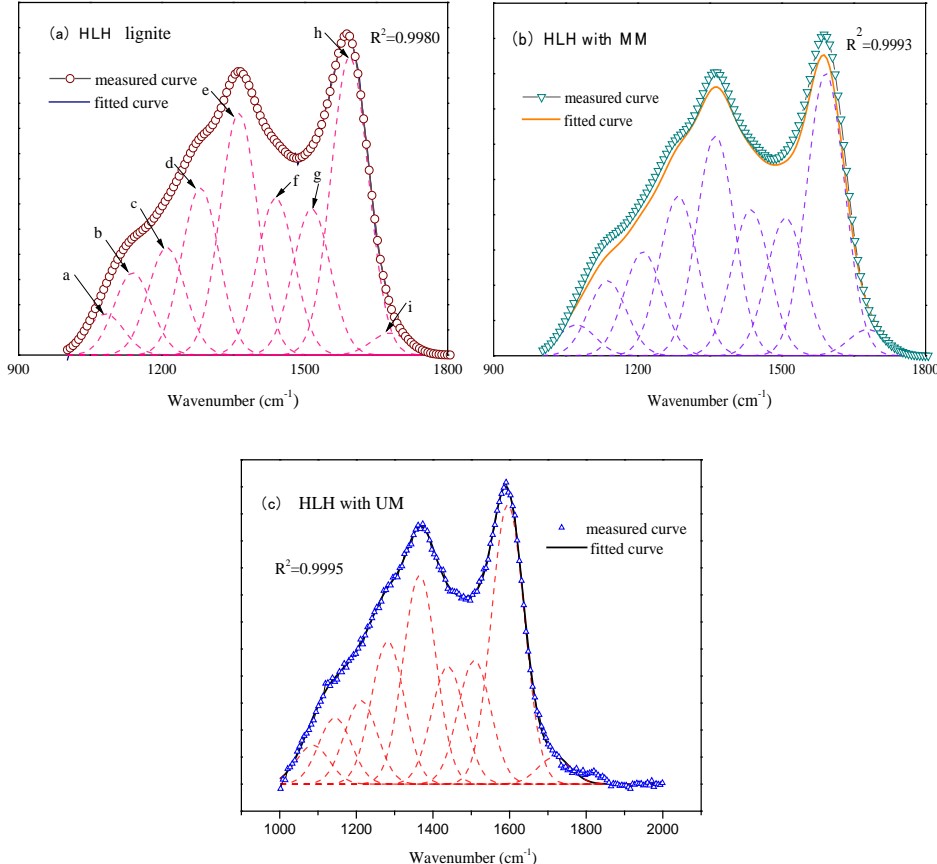

**Figure 6.** Raman peak separation fitting charts of HLH coal samples before and after reformation.

**Table 10.** Raman spectrum structural parameters of different coal samples.

| Sample | Peak Position | | Peak Area | | Peak Height | | $P_{G-D}$ | $A_D/A_G$ | $I_D/I_G$ |
|---|---|---|---|---|---|---|---|---|---|
| | $P_D$ | $P_G$ | $A_D$ | $A_G$ | $I_D$ | $I_G$ | | | |
| HLH | 1361.1 | 1594.3 | 47,030 | 57,990 | 4782.3 | 5926.3 | 233.2 | 0.81 | 0.81 |
| MM | 1362.1 | 1591.8 | 35,770 | 45,858 | 3105.5 | 3997.8 | 229.7 | 0.78 | 0.78 |
| UM | 1365.3 | 1595.6 | 41,480 | 55,500 | 3867.6 | 5173.1 | 230.3 | 0.75 | 0.75 |

From the $A_D$ and $A_G$ values of coal samples before and after modification, it can be found that the D peak area of coal samples after microwave and ultrasonic treatment decreases, especially the D peak area of coal samples after microwave treatment decreases significantly, which also shows that microwave treatment makes HLH structure more complete. Compared with G peak area $A_G$, the $A_G$ value of HLH structure after microwave modification is significantly smaller than that of HLH. It can be seen that the total number of aromatic rings in macromolecular structure of HLH treated by microwave is the smallest, and the enrichment degree of aromatic carbon is the lowest, followed by the macromolecular structure of HLH coal treated by ultrasound, while the content of aromatic rings in the macromolecular structure of HLH coal is the highest and the enrichment degree of aromatic carbon is the largest.

$I_D/I_G$ is usually used to evaluate the disordering degree of carbon materials, and it decreases with the increase of graphitization degree. After microwave treatment, the $I_D/I_G$ value of HLH was reduced from 0.81 to 0.78. Similarly, after ultrasonic treatment, the $I_D/I_G$ value of HLH was also reduced to 0.75, and the $I_D/I_G$ value was reduced. This indicated that the ordering degree of aromatic ring layers in the structure increased and the content of fat chain and side chain decreased after microwave and ultrasonic treatment.

### 3. HLH Molecular Model Before and After Modification

*3.1. Determination of the Type and Number of Aromatic Ring*

The average molecular formula of HLH is $C_{167}N_3O_{27}H_{149}$ by elemental analysis. The average molecular formula of microwave modified HLH is $C_{148}H_{129}N_3O_{20}$, and that of ultrasonic modified HLH is $C_{155}H_{131}O_{23}N_{23}$. Combined with XPS and elemental analysis, it is found that S content in HLH lignite is extremely small. S element was added to the macromolecular model, but it was found that the percentage of S atom in the analysis of experimental elements was about 1.5%, which was obviously inconsistent with the actual results. Therefore, a small amount of S content was neglected in the construction of the macromolecular model of HLH.

$X_{BP}$ is calculated by using the twelve structural parameters, which is calculated by $^{13}$C NMR. The ratio of aromatic bridge carbon to periphery carbon of HLH raw coal is 0.26, that of microwave modified HLH is 0.29, and that of ultrasonic modified HLH is 0.28. The $X_{BP}$ of naphthalene ring with two rings is 0.25, and that of anthracene ring with three rings is 0.40. Therefore, the aromatic framework of HLH raw coal structure model is mainly composed of benzene ring and naphthalene ring. After HLH with microwave modification model and HLH with ultrasonic modification aromatic framework is mainly composed of naphthalene ring, with anthracene ring and benzene ring supplemented. In order to make the $X_{BP}$ of HLH raw coal and HLH model after microwave and ultrasonic modification close to 0.26, 0.29 and 0.28, the combination number of benzene, naphthalene and anthracene rings in its structural model needs to be adjusted continuously. The type and number of aromatic rings in the three structural models are determined. The results are shown in Table 11.

**Table 11.** Type and quantities of aromatic rings of HLH before and after modification.

| Type of Aromatic Unit Structure | Number | | |
|---|---|---|---|
| | HLH | MM | UM |
|  | 6 | 4 | 5 |
|  | 5 | 3 | 3 |
|  | 2 | 2 | 2 |
|  | 1 | 1 | 1 |
|  | — | 1 | 1 |

Comparing the previous $^{13}$C NMR and FTIR spectra, it is found that the proportion of oxygen bonded by carbon-oxygen double bond is much smaller than that bonded by carbon-oxygen single bond, which is consistent with the XPS test results. This shows that the main forms of oxygen in macromolecular structure before and after HLH modification are mostly in the form of ether bond and hydroxyl bond of carbon-oxygen single bond, and the other oxygen-containing structures are in the second place. The final form of oxygen in the structure is determined by constantly adjusting the structure of oxygen.

The main forms of nitrogen in HLH macromolecular structure are pyridine nitrogen and pyrrole nitrogen. According to XPS analysis, the number of nitrogen atoms is always 3 of HLH before and after

modification, and the main forms of nitrogen elements were pyridine nitrogen and pyrrole nitrogen the content ratio was 2:1. Therefore, two pyridine rings and one pyrrole ring were added to the HLH structure model before and after modification.

### 3.2. Model Construction

Wiser coal chemical model is used as the basis, which is widely accepted and considered to be comprehensive and reasonable. Combining with the above calculation results, the existing forms and proportions of each part of the macromolecular structure model before and after modification of HLH are summarized and analyzed. Finally, the two-dimensional molecular model of HLH before and after modification is preliminarily established according to the chemical structure characteristics obtained from the experiments. As shown in Figures 7–9.

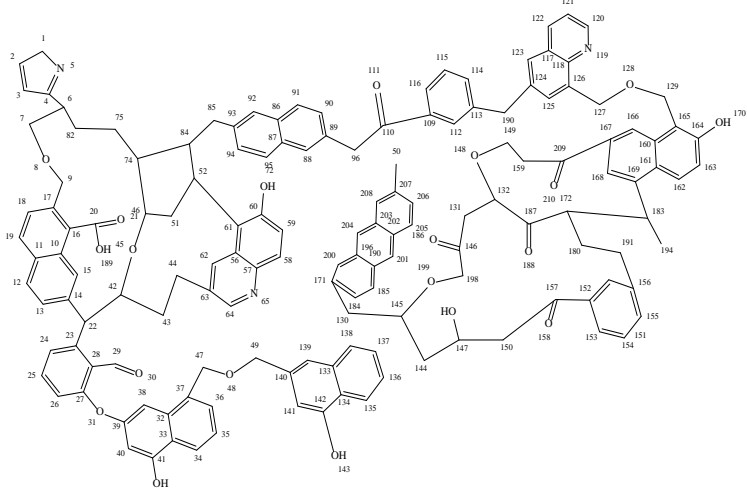

**Figure 7.** Final two-dimensional model macromolecular structure diagram of HLH.

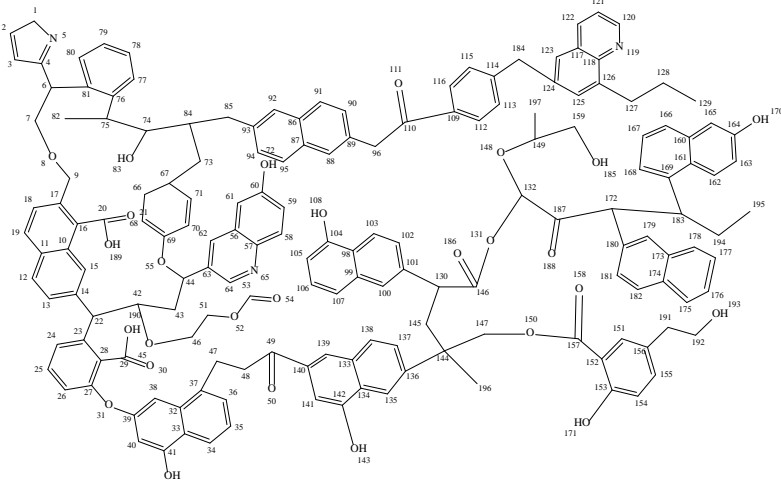

**Figure 8.** Final two-dimensional model macromolecular structure of HLH after microwave modification.

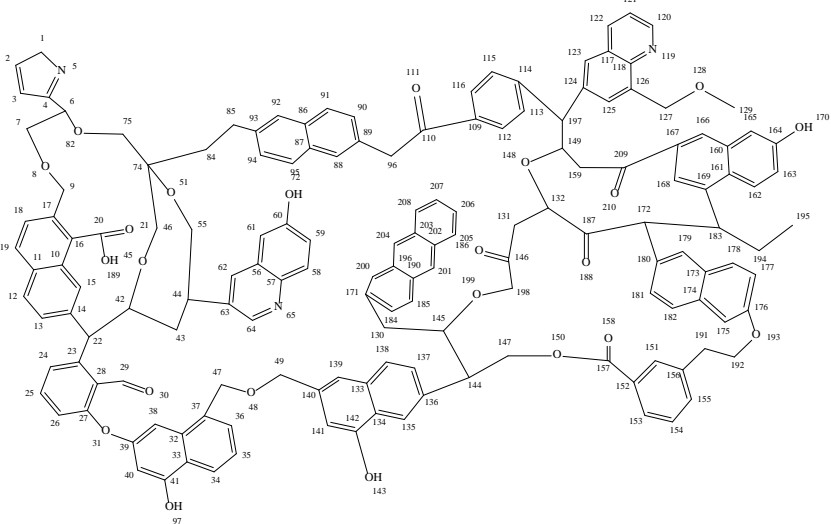

**Figure 9.** Final two-dimensional model macromolecular structure of HLH after Ultrasonic modification.

### 3.3. Verification of Model

The [13]C NMR chemical shifts of the three models were calculated and compared with the experimental [13]C NMR chemical shifts. Because of the complexity and diversity of coal's macromolecule structure, it is necessary to constantly adjust the types and quantities of its structural units in the construction process, so as to make its [13]C NMR simulation spectra better consistent with the experimental spectra. The comparison of [13]C NMR simulation spectra and experimental spectra of the three models is shown in Figure 10.

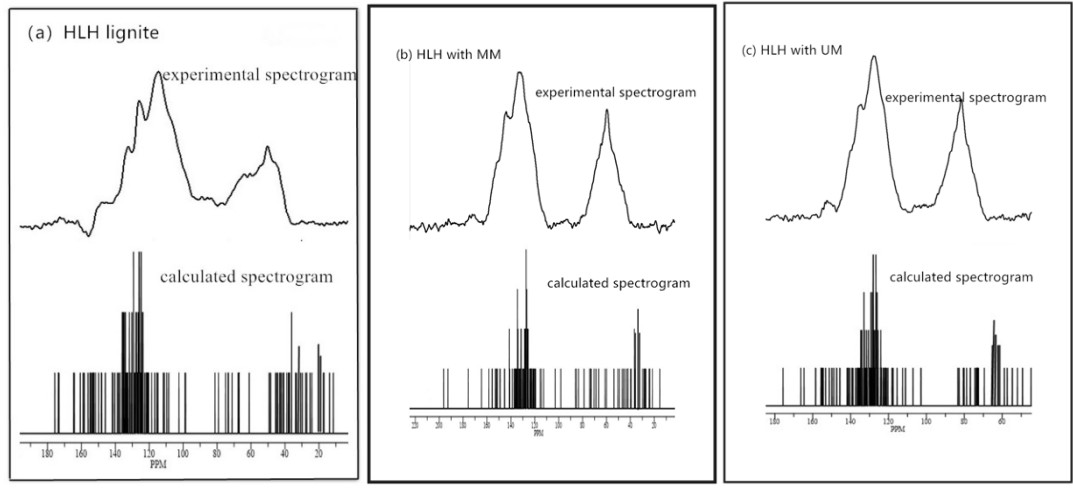

**Figure 10.** Comparison of [13]C NMR computational spectra and experimental spectra of HLH final two-dimensional model before and after modification: (**a**) HLH, (**b**) HLH after microwave modification, (**c**) HLH after Ultrasonic modification.

It can be seen that the [13]C NMR simulation spectra of the three models are in good agreement with the experimental spectra. There are some errors between the two spectra due to some uncontrollable factors in the experimental process, but it is considered acceptable and will not affect the characterization of average macromolecular structure of HLH coal samples.

In order to adjust the structure of the model to approximate the experimental data, most researchers adopted the simulation of [13]C NMR spectrum, but which cannot avoid the choice of isomers [23,24]. At the same time, the model obtained by this method is only a conceptual structure, which cannot reflect the properties of chemical reactions. By using the main covalent bond concentration instead of

the $^{13}$C NMR simulation spectrum, the molecular model can be better corrected. According to the 11 structural parameters ($f_a$, $f_a^H$, $f_a^O$, $f_a^B$, $f_a^S$, $f_a^C$, $f_{al}$, $f_{al}^A$, $f_{al}^M$, $f_{al}^H$, $f_{al}^O$) obtained from $^{13}$C NMB data, nine covalent bond concentrations of HLH lignite (Car-Car, Car-Cal, Cal-Cal, Car-H, Cal-H, Car-O, Cal-O, Cal=O, and O-H.) can be obtained. This covalent concentration method reflects the essence of $^{13}$C NMR simulation spectroscopy. By comparing the simulated concentration of the main covalent bond with the experimental results, the preliminary two-dimensional molecular model is modified. The concentrations of nine types covalent bonds in coal can be determined by Equations (7)–(15).

$$con_{C_a-C_a} = \frac{1}{2}\left[\frac{C\%}{12}\left(3f_a - f_a^H - f_a^S - f_a^O\right)\right] \tag{7}$$

$$con_{C_a-C_{al}} = \frac{C\%}{12}f_a^S \tag{8}$$

$$\begin{aligned} con_{C_{al}-C_{al}} &= \frac{1}{2}\left[\frac{C\%}{12}\left(4f_{al} + 2f_a^C - f_a^C - f_{al}^O\right) - n_{C_{al}-H}\right] \\ &= -\frac{H\%}{2} + \frac{O\%}{16} + \frac{1}{2}\left[\frac{C\%}{12}\left(4f_{al} + f_a^H - f_a^S - f_a^O - 2f_{al}^O - f_a^C\right)\right] \end{aligned} \tag{9}$$

$$con_{C_a-H} = \frac{C\%}{12}f_a^H \tag{10}$$

$$\begin{aligned} con_{C_{al}-H} &= H\% - con_{O-H} - con_{C_a-H} \\ &= H\% - 2\frac{O\%}{16} + \frac{C\%}{12}\left(f_a^O + f_{al}^O + 2f_a^O + 3f_a^C - f_{ar}^H\right) \end{aligned} \tag{11}$$

$$con_{C_{ar}-O} = \frac{C\%}{12}f_{ar}^O \tag{12}$$

$$con_{C_{al}-O} = \frac{C\%}{12}\left(f_{al}^O + f_a^C\right) \tag{13}$$

$$con_{C_{al=o}} = \frac{C\%}{12}\left(f_a^C\right) \tag{14}$$

$$con_{O-H} = \frac{2O\%}{16} - \frac{C\%}{12}\left(f_a^O + f_{al}^O + 3f_a^C\right) \tag{15}$$

Based on the formula, the information of covalent bond concentration of the three final planar model structures is calculated and summarized in Table 12, in order to more intuitively compare and analyze the difference between the covalent bond concentration obtained from the experiment and that calculated from the model. Then, the molecular models of the three structures obtained from HLH raw coal, microwave, and ultrasonic upgrading of HLH coal, respectively, were adjusted.

**Table 12.** Covalent bond concentrations of 3 structural models of HLH before and after modification.

| Sample | $con_{Ca-Ca}$ | $con_{Ca-Cal}$ | $con_{Cal-Cal}$ | $con_{Ca-H}$ | $con_{Cal-H}$ | $con_{Ca-O}$ | $con_{Cal-O}$ | $con_{Cal=O}$ | $con_{O-H}$ |
|---|---|---|---|---|---|---|---|---|---|
| HLH | 46.701 | 6.943 | 18.037 | 16.410 | 35.768 | 5.303 | 5.389 | 2.828 | 7.208 |
| HLH Model | 46.233 | 6.822 | 17.354 | 17.231 | 35.154 | 5.036 | 5.673. | 2.32 | 7.053 |
| MM | 47.669 | 10.053 | 19.488 | 14.499 | 36.133 | 4.851 | 3.621 | 2.113 | 5.252 |
| MM Model | 47.276 | 10.997 | 17.376 | 15.547 | 34.453 | 4.784 | 3.435 | 2.349 | 5.127 |
| UM | 47.095 | 9.934 | 20.374 | 15.093 | 38.309 | 4.793 | 2.133 | 2.005 | 6.407 |
| UM Model | 46.967 | 9.322 | 19.763 | 15.216 | 37.867 | 4.452 | 2.243 | 2.105 | 6.395 |

According to the covalent bond concentration obtained from the experiment and the concentration calculated by the model, it was found that the concentration of the nine covalent bonds is not different from that calculated by the model, but the difference between the experimental concentration of $con_{Cal-Cal}$ and $con_{Car-H}$ and the calculated concentration of the model is relatively large. The reason for this may be that there is a certain difference between the proportion of elements in the model and that in the experimental measurement, which makes the calculation of the nine covalent bonds concentration more intensive.

*3.4. Adjustment of Model*

After the final model of molecular structure is obtained, the molecular formulas and element contents of the three models of HLH under different modification conditions are calculated. By comparing the results of Tables 13 and 14, it can be seen that the element contents of the three models are very close to the experimental values, and there is no significant difference, which verifies the reliability of the model.

**Table 13.** Element content measured by experiments of HLH before and after modification.

| Sample | Ultimate Analysis (wt. %), ad | | | |
|--------|------|------|-------|------|
|        | C    | H    | O     | N    |
| HLH    | 78.56 | 6.36 | 13.48 | 1.61 |
| MM     | 80.87 | 5.99 | 11.39 | 1.76 |
| UM     | 81.97 | 5.67 | 10.41 | 1.95 |

**Table 14.** Molecular formula and element content of the model of HLH before and after modification.

| Sample | Molecular Formula | Ultimate Analysis (wt. %) | | | |
|--------|-------------------|-------|------|-------|------|
|        |                   | C     | H    | O     | N    |
| HLH    | $C_{167}H_{151}N_3O_{27}$ | 76.21 | 5.78 | 16.41 | 1.6  |
| MM     | $C_{148}H_{129}N_3O_{20}$ | 78.52 | 5.73 | 14.10 | 1.85 |
| UM     | $C_{155}H_{131}N_3O_{23}$ | 77.45 | 5.49 | 15.31 | 1.75 |

*3.5. Construction of 3- Dimensional Model*

The three-dimensional model is constructed for the 2- dimensional structure of HLH before and after modification, whose three-dimensional geometric optimization configuration is calculated by MM and MD of Forcite module in Materials Studio 8.0 software. The structure model of HLH before and after optimization is shown in Figures 11–13, and the energy change in the process of optimization is shown in Table 15.

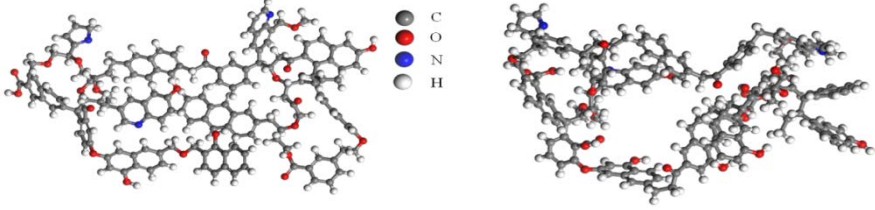

(**a**) Initial 3-dimensional structure model    (**b**) Optimized by MM and MD

**Figure 11.** 3-dimensional structure of HLH lignite model before and after geometric optimization.

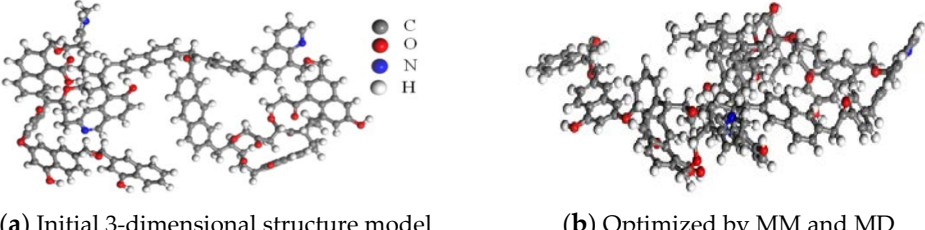

(**a**) Initial 3-dimensional structure model    (**b**) Optimized by MM and MD

**Figure 12.** 3-dimensional structure of HLH lignite model modified by microwave before and after geometric optimization.

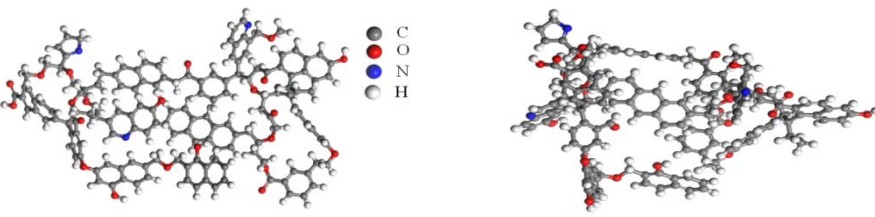

(**a**) Initial 3-dimensional structure model (**b**) Optimized by MM and MD

**Figure 13.** 3-dimensional structure of HLH lignite model modified by ultrasound before and after geometric optimization.

**Table 15.** Energy comparison before and after HLH structural model optimization.

| Sample | | Total Energy (Kcal·mol$^{-1}$) | Valence Energy (Kcal·mol$^{-1}$) | | | | Non-Bond Energy (Kcal·mol$^{-1}$) | | |
|---|---|---|---|---|---|---|---|---|---|
| | | | $E_B$ | $E_A$ | $E_T$ | $E_I$ | $E_H$ | $E_{van}$ | $E_E$ |
| HLH | Initial | 6030.56 | 2602.16 | 65.21 | 90.59 | 2.05 | 0 | 3270.55 | 0 |
| | Final | 810.65 | 204.89 | 197.06 | 119.08 | 18.42 | 0 | 342.05 | −70.85 |
| MM | Initial | 6310.80 | 2225.16 | 153.04 | 89.24 | 1.68 | 0 | 3841.68 | 0 |
| | Final | 850.20 | 191.04 | 217.58 | 139.81 | 24.99 | 0 | 344.30 | −67.52 |
| UM | Initial | 7608.20. | 2402.78 | 142.34 | 122.83 | 1.56 | 0 | 4938.69 | 0 |
| | Final | 846.37 | 187.50 | 189.76 | 134.07 | 17.57 | 0 | 348.01 | −30.54 |

It can be seen from this that the total energy of the minimum energy structures of the three structures decreases sharply. Compared with other terms, the value of Van der Waals energy ($E_{Van}$) in the non-bonding energy is the largest, which constitutes the most important part of the potential energy. Therefore, the decrease of the inter-molecular Van der Waals energy ($E_{Van}$) is also the main factor that makes the HLH macromolecular structure model stable in space.

## 4. Conclusions

(1) The molecular formula of the three structures was determined by elemental analysis. The average molecular formula of HLH raw coal configuration was $C_{167}N_3O_{27}H_{149}$, and the aromatic part consists of five benzene rings, six naphthalene rings, two pyrrole rings and one pyridine ring. The average molecular formula of Mm coal configuration was $C_{148}H_{129}N_3O_{20}$, the aromatic structure includes three benzene rings, four anthracene rings, four naphthalene rings, one anthracene ring, one pyridine ring and two pyrrole rings. And the average molecular formula of Um coal configuration was $C_{155}H_{131}O_{23}N_{23}$, the aromatic structure includes three benzene rings, five anthracene rings, four naphthalene rings, one anthracene ring, one pyridine ring, and two pyrrole rings.

(2) It was found that after microwave and ultrasonic treatment, the orderliness of aromatic ring layer arrangement increased. The content of fat chain and side chain decreased, and the existence form of oxygen atoms also changed, in which the proportion of C-O and COO- form shows a decreasing trend. It was found that the total energy of the three structures decreased significantly after optimization. The chemical bonds between the atoms are obviously bent and distorted, and the space configuration is more stereoscopic. Although the model constructed in this paper is not the most comprehensive and optimized configuration before and after HLH modification, some problems such as isomers are considered in the process of construction, which has a certain reference value for better understanding and application of lignite upgrading.

**Author Contributions:** J.L. provided methodology and original draft writing. Z.H. and Y.Y. conceived and designed the study. S.Q. performed the experiments. Y.Y. provided the samples. S.Q. wrote the paper. Z.H. reviewed and edited the manuscript. All authors have read and agreed to the published version of the manuscript.

**Funding:** This work was financially supported by National Natural Science Foundation of China (51874171) and and University of Science and Technology Liaoning Talent Project Grants (No. 601011507-05).

**Conflicts of Interest:** The authors declared that they have no conflict of interest to this work. We declare that we do not have any commercial or associative interest that represents a conflict of interest in connection with the work submitted.

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
