# Peer review of "Experiments and 3D Molecular Model Construction of Lignite under Different Modification Treatment"

_processes, doi:10.3390/pr8040399_

Round 1

Reviewer 1 Report

Dear editor,

the submitted manuscript is a gorgeous work using multiple technics and analytical procedures to put light on the structural differences of lignite molecule under different treatment. Most important is the application of this research to the industry.

Two minor comments that in my opinion would improve this work are listed below.

  1. Abstract: Please add the key results of the methods that were used for this study. 
  2. 2.2.1. FTIR results analysis (line 68): Please add the coresponding literature of the assignments in the tables 2 to 5.

I look forward seeing this very interesting study published.

Reviewer 2 Report

In this manuscript titled "3D Molecular Model Construction: Huolinhe Lignite with Different Modification Treatment", the authors attempt to propose the 3D structure of Huolinhe Lignite and its altered structure after the microwave or ultrasonic treatment. I recommend this manuscript could be published because the results provide a new research perspective for the lignite and its utilization. But I recommend this manuscript should be amended further for the following reasons:

  1. In section 2.1, the preparations for the three kinds of samples are not clearly stated. It is also little vague whether the HLH samples were pulverized and processed with water. The detailed information is crucial to the final conclusions.
  2. The 3D structure of Huolinhe Lignite has been investigated in the following publications:

Yang, F., et al., A new insight into the structure of Huolinhe lignite based on the yields of benzene carboxylic acids, Fuel, 2017, Vol 189, pp 408-418. (http://doi.org/10.1016/j.fuel.2016.10.112); Xu, F., et al., Construction and evaluation of chemical structure model of Huolinhe lignite using molecular modeling, RSC Advances, 2017, Vol 7, pp41512-41519. (http://doi.org/10.1039/C7RA07387A).

The author's literature review needs to be further enriched and improved, highlighting the differences and novelties between this work and the predecessors.

  1. The full-names of the abbreviations like Mad, Aad, FCsd, and Vdaf in Table 1 should be given.
  2. Due to the dissolution of minerals in the coal by water, I wonder whether the chemical elements in the coal vary depending on the degree of water treatment. Are the three samples all the same degree? It needs to be explained in details.
  3. There are several small typos and grammatical errors in this manuscript and its reference section. The authors should pay more attention.
